# ICU-Acquired Hypernatremia Is Associated with Persistent Inflammation, Immunosuppression and Catabolism Syndrome

**DOI:** 10.3390/jcm9093017

**Published:** 2020-09-18

**Authors:** Christopher Rugg, Mathias Ströhle, Benedikt Treml, Mirjam Bachler, Stefan Schmid, Janett Kreutziger

**Affiliations:** 1Department of General and Surgical Critical Care Medicine, Innsbruck Medical University Hospital, Anichstr. 35, 6020 Innsbruck, Austria; mathias.stroehle@tirol-kliniken.at (M.S.); benedikt.treml@tirol-kliniken.at (B.T.); mirjam.bachler@tirol-kliniken.at (M.B.); stefan.schmid@tirol-kliniken.at (S.S.); janett.kreutziger@tirol-kliniken.at (J.K.); 2Institute for Sports Medicine, Alpine Medicine and Health Tourism, UMIT—University for Health Sciences, Medical Informatics and Technology, 6060 Hall in Tirol, Austria

**Keywords:** hypernatremia, PICS, catabolism, inflammation, urea-to-creatinine ratio

## Abstract

Developing hypernatremia while on intensive care unit (ICU) is a common problem with various undesirable effects. A link to persistent inflammation, immunosuppression and catabolism syndrome (PICS) can be established in two ways. On the one hand, hypernatremia can lead to inflammation and catabolism via hyperosmolar cell stress, and on the other, profound catabolism can lead to hypernatremia via urea-induced osmotic diuresis. In this retrospective single-center study, we examined 115 patients with prolonged ICU stays (≥14 days) and sufficient renal function. Depending on their serum sodium concentrations between ICU day 7 and 21, allocation to a hypernatremic (high) and a nonhypernatremic group (low) took place. Distinct signs of PICS were detectable within the complete cohort. Thirty-three of them (28.7%) suffered from ICU-acquired hypernatremia, which was associated with explicitly higher signs of inflammation and ongoing catabolism as well as a prolonged ICU length of stay. Catabolism was discriminated better by the urea generation rate and the urea-to-creatinine ratio than by serum albumin concentration. An assignable cause for hypernatremia was the urea-induced osmotic diuresis. When dealing with ICU patients requiring prolonged treatment, hypernatremia should at least trigger thoughts on PICS as a contributing factor. In this regard, the urea-to-creatinine ratio is an easily accessible biomarker for catabolism.

## 1. Introduction

Hypernatremia is a common problem in critically ill patients [1]. While general causes are simply broken down to a net gain in total sodium or a net loss of free water, clinical discrimination of the main underlying pathology is not always easily done. The deleterious effects of this hyperosmolar state range from neurological pathologies (brain cell shrinkage, restlessness, coma, cerebral demyelination) [2,3] to possible rhabdomyolysis [4,5], disturbed glucose utilization as well as impaired gluconeogenesis [6,7] and decreased left ventricular function [8]. Depending on the type of intensive care unit (ICU), 2–6% of patients are hypernatremic on admission and up to 7–26% develop hypernatremia during treatment—predominantly within the first week [9,10,11,12,13,14]. Regardless of whether it was present at admission or acquired during intensive care, hypernatremia has been shown to be a predictor and independent risk factor for mortality by numerous studies in various situations [9,12,13,15,16].

Another ICU-acquired disorder associated with an undesirable outcome is the persistent inflammation, immunosuppression and catabolism syndrome (PICS)—not to be mistaken for the so-called postintensive care syndrome, which is abbreviated likewise. Since its first description by Gentile et al. in 2012 [17], the concept of PICS has been validated and is becoming more and more accepted as the underlying pathophysiology of chronic critical illness (CCI) [18,19,20,21,22,23,24]. The paradigm implies that, following the simultaneously triggered pro- and anti-inflammatory responses to a major inflammatory insult (e.g., trauma, burns, sepsis, acute pancreatitis, etc.), the increasing number of acute survivors either proceed to a fairly rapid recovery or a prolonged trajectory partially ending in CCI [18,19]. Clinically speaking, these patients present with a prolonged ICU stay (typically >14 days) under the coexistence of ongoing inflammation and immunosuppression, resulting in persistent catabolism and organ dysfunction [18,19,20]. Besides an ICU length of stay (LOS) of over 14 days, applied clinical markers defining the diagnosis are as follows: a C-reactive protein (CRP) over 0.15 mg/dL as sign of inflammation; a total lymphocyte count under 0.800 G/L as sign of immunosuppression; and a serum albumin concentration under 3.0 g/dL, a creatinine height index under 80% or a weight loss over 10% as signs of ongoing catabolism [17]. Regarding sepsis patients on surgical ICU, an observational study was able to show that more than half of the acute survivors ended up developing CCI. This was associated with older age, an increased rate of hospital-acquired infections and a six-month survival of merely 63% [25].

The link back to hypernatremia is theoretically done in two ways. On the one hand, persistent catabolism with elevated ureagenesis can lead to profuse urine urea output and subsequently to urea-induced osmotic diuresis with loss of electrolyte-free water and resulting hypernatremia [26,27,28,29,30,31]. On the other hand, hypernatremia itself can again promote protein catabolism and also systemic inflammation, primarily via hyperosmolar cell stress [32,33,34,35]. We therefore hypothesized that in patients with prolonged ICU LOS but sufficient renal function, hypernatremia is associated with PICS via chronic inflammation and excessive catabolism. The aim of this study was to elucidate this association by analyzing patients with a prolonged stay in ICU and therefore existing susceptibility for developing PICS. We assumed a link between hypernatremia and PICS to be partially due to urea-induced osmotic diuresis. Differences between hypernatremic and nonhypernatremic patients was analyzed, especially for signs of inflammation and excessive catabolism as well as the loss of electrolyte-free water as assignable cause for hypernatremia.

## 2. Materials and Methods

This retrospective study was approved by the Ethics Committee of the Medical University of Innsbruck (EK Nr.: 1011/2020) and the Institutional Review Board. Due to its retrospective design, a consent to participate was not applicable.

The study was conducted at the Department for General and Surgical Critical Care Medicine of Innsbruck Medical University Hospital. Containing 23 level 3 beds, our department annually treats 650–700 multiple trauma and burn injury patients as well as patients following scheduled and emergency cardiac, vascular, thoracic and abdominal surgery and transplantations. The primary study population consisted of patients treated at our department within a 2-year timeframe from 1 January 2017, to 31 December 2019. Aiming for a subset with possible signs of PICS, we decided to define our main observation period as ICU day 7 to 21. From *n* = 1374 eligible patients, *n* = 1052 were excluded due to an ICU LOS of less than 14 days (Figure A1). In order to properly analyze hypernatremia and certain signs of catabolism (urea-to-creatinine ratio, urea generation rate), a sufficient renal function during the main observation period was required. Due to missing data on daily creatinine clearances, *n* = 169 patients were excluded. Another *n* = 38 patients were excluded due to a mean creatinine clearance of less than 45 mL/min during ICU day 7 to 21. The remaining *n* = 115 patients were divided into a high and a low group depending on the mean of their daily mean serum sodium concentrations (Na_S_) as measured by direct potentiometry in blood gas analysis during the main observation period. The cut-off was set as 145 mmol/L and resulted in *n* = 33 patients in the high and *n* = 82 patients in the low Na_S_ group. Inclusion criteria as well as steps of exclusion and group division are illustrated as a flowchart in Figure A1.

Medical information was obtained from the local hospital information system as well as the ICU patient data management system (PDMS; GE, Centricity Critical Care 9.0 SP1). Extracted parameters were age, weight, Simplified Acute Physiology Score on admission (SAPS-III) [36,37] and ICU LOS, Na_S_ as assessed by direct potentiometry, plasma osmolality (Osmo_Pl_), fluid balance, urine output (OUT_Ur_), serum creatinine (Crea_S_) and urea (Urea_S_), serum glucose concentration (Gluc_S_), 24 h urine lab (sodium- (Na_Ur_), potassium- (K_Ur_), creatinine- (Crea_Ur_), urea concentration (Urea_Ur_), osmolality (Osmo_Ur_)), CRP, procalcitonin (PCT), leucocytes and total lymphocytes. Derived or calculated variables included creatinine- (CrCl), free water- (FWC) and electrolyte-free water clearance (EFWC), rate of urea generation (UreaGR), serum urea-to-creatinine ratio (both as mg/dL) and ratio of urine osmolality due to urea vs. nonurea osmoles (Osmo_Urea_/Osmo_Rest_). Renal protein loss was deduced from total urea excretion by deriving nitrogen loss and approximating a nitrogen proportion of 16% in a balanced protein mix. The patient’s total sodium intake was generated by our PDMS, where sodium content of IV fluids and drugs are stored. Urea generation rate was defined as the sum of total amount of renal urea excretion and the amount of serum urea changes. The latter was estimated by multiplying changes in serum urea concentration with expected total body water (0.6 × body weight). Free and modified electrolyte-free water clearances were calculated according to the formulas described by Nguyen and Kurtz in 2012 (Table 1) [38].

After division in a hyper- (high) and nonhypernatremic (low) group, comparison was conducted regarding demographics (Table 2) and clinical signs of PICS (Figure 1, Table 3). Inflammatory and immunosuppressive parameters included CRP, PCT, leucocyte and absolute lymphocyte count. Due to scarcity of total measurements, absolute lymphocyte count was presented as one average per patient during ICU day 7–21. Surrogate parameters for ongoing catabolism included serum albumin concentration, urea generation rate and serum urea-to-creatinine ratio. An elevated urea-to-creatinine ratio above 75 (75 mg/dL:mg/dL = 141 mmol/L:mmol/L) has recently been published as a novel biomarker for critical illness-associated catabolism [39]. Confirmation as a marker for hypermetabolism in severely burned patients has also been done by our group [26]. In the next step, fluid balance, sodium intake, renal function, urine output, urine sodium output and osmole composition were compared in order to elucidate a possible pathophysiologic link between hypernatremia and PICS (Figure 2, Table 3).

Utilizing R (R Core Team) and RStudio version 1.2.5001 (RStudio, Inc., Boston, MA, USA), statistical analysis of group differences was performed for the main observation period (ICU day 7–21). Figure 1 and Figure 2 also present a graphical analysis of selected parameters from ICU admission to day 21. Data were tested for normal distribution by Shapiro–Francia test and presented as median and interquartile range (Q1–Q3) due to non-normal distribution. In this nonparametric setting, the two independent groups consisting of multiple, intraindividual longitudinal data were analyzed using a modified ANOVA-type statistic (R package: nparLD) [40]. Regarding demographics, chi-squared test was performed to detect group differences in frequencies and the Mann–Whitney *U*-test for group differences of continuous data (Table 2 and absolute lymphocytes in Table 3). *p* < 0.05 was considered significant.

## 3. Results

### 3.1. Demographics

Studying ICU patients with sufficient renal function during ICU day 7–21 and concomitantly dividing them by mean Na_S_ yielded one hypernatremic group (high; *n* = 33) and one nonhypernatremic group (low; *n* = 82). Both groups were similar regarding age, gender distribution, admission type and SAPS III admission score. Although ICU outcome was identical in both groups, ICU LOS differed significantly. Median LOS was four days longer in the high group compared to the low group (Table 2).

### 3.2. Association between Hypernatremia and PICS

The complete study population as well as both groups were separately analyzed with respect to signs of PICS (Table 3). With elevated CRP, urea generation rate and urea-to-creatinine values as well as decreased lymphocyte counts and serum albumin concentrations, the study population in total presented with clear signs of PICS. The high group, however, presented with explicitly increased PICS signs compared to the low group. Leucocytes, CRP, PCT, urea generation rate and urea-to-creatinine ratio were significantly higher. Serum albumin concentrations did not differ but were rather low in both groups. The median of absolute lymphocyte counts was clearly lower in the high group but did not reach clear significance, mainly due to a high variance in both groups. While Table 3 presents median values over the complete observation timeframe (ICU day 7–21), Figure 1 presents a graphical day-by-day analysis from admission to ICU day 21.

### 3.3. Assignable Causes for Hypernatremia

In order to elucidate a possible interrelation between hypernatremia and PICS, both groups were analyzed with regard to urea-induced osmotic diuresis and subsequent loss of electrolyte-free water. Furthermore, fluid balances, sodium intake and general renal function were compared. As shown in Figure 2, the increased urea generation rate in the high group was also accompanied by a raised fraction of urea in urine osmole excretion. The combination of a consecutively reduced urine sodium concentration and a higher urine output was what defined osmotic diuresis and led to a significantly elevated EFWC in the high group. Creatinine clearances, as measured directly by 24-h urine sampling, were higher in the low group but sufficiently high in both groups (Table 3). Proving adequate renal function, free water clearances—representing renal concentration ability—did not differ between the groups and were highly negative in both groups. While fluid balances were significantly higher, total sodium input was higher but just missed significance in the high group. Median sodium balances, as difference between sodium input and renal sodium output, were negative for both groups during the observation period (−13 mmol/d in the high and −106 mmol/d in the low group).

## 4. Discussion

Distinct clinical signs of PICS were detectable within the complete cohort of 115 ICU patients with a prolonged LOS (≥14 days) but sufficient renal function. Thirty-three of them (28.7%) suffered from ICU-acquired hypernatremia. This was associated with explicitly higher signs of inflammation and ongoing catabolism as well as prolonged ICU LOS. Catabolism was discriminated better by the urea generation rate and the urea-to-creatinine ratio than by serum albumin concentration. One assignable cause for hypernatremia in these patients is urea-induced osmotic diuresis, subsequently causing loss of electrolyte-free water. When dealing with critically ill patients requiring prolonged ICU treatment, hypernatremia under sufficient renal function should at least trigger thoughts on PICS as a possible contributor. In this regard, the urea-to-creatinine ratio is an easily accessible biomarker for catabolism.

### 4.1. Demographics

Depending on the type of ICU (surgical, medical or mixed), the incidence of ICU-acquired hypernatremia has been described in ranges from 7 to as much as 26% [9,11,12,14]. A partially differing cut-off defining hypernatremia (145 or 150 mmol/L) may explain one part of this high variability, but center-specific treatment regimens are certainly also of importance. Furthermore, inclusion criteria itself can clearly contribute to the observed discrepancies. In this regard, we must state an even higher incidence of 28.7% in our preselected population with prolonged LOS on a mainly surgical ICU. We assume that at least part of this higher incidence is explained just by profuse catabolism as also shown in a study on severely burned, hypermetabolic patients in our ICU [26]. In contrast to the existing literature, no impact on mortality was able to be shown for hypernatremic patients in this study. Merely 6.1% of our study population had a fatal outcome, which is quite contrary to other studies, where mortality for ICU-acquired hypernatremic patients reached up to over 40% [1,9,11,12,14]. The exclusion of patients with insufficient renal function or requirement of renal replacement therapy, of course, partially excludes more severely ill patients and therefore influences mortality rates. A prolonged ICU LOS for patients suffering from ICU-acquired hypernatremia as shown in this study is, however, in accordance with recent literature and emphasizes the significance and importance of the observed illness [9,11,12,41]. Yet, whether hypernatremia, inflammation, catabolism or the combination of all three led to the observed prolonged ICU LOS cannot be determined by the presented results in this study.

### 4.2. Association between Hypernatremia and PICS

Combining new immunological insights with great clinical experience is how PICS was first described in 2012 [17]. The attempt to translate fundamental research into readily available surrogates in order to describe a clinical condition is how biomarkers and their cut-offs were determined. There is still no consensus on the exact definition of chronic critical illness, but a LOS past ICU day 14—as also used in this study—is often accepted [18,21,22]. The transition point to a beginning PICS potentially ending in CCI seems to be between ICU day 7 and 10 [17,18], hence the definition of the main observation period in this study. Regarding markers of inflammation, CRP levels were clearly within the PICS definition, even when utilizing a recently published threshold of 3 mg/dL derived from a retrospective study on CRP clustering in PICS [42]. Immunosuppression, however, is not so easily proven. While fundamental research on pathophysiologic processes are increasing [43], easily accessible surrogates are hard to define. The number of absolute lymphocyte measurements conducted in this study was rather scarce, and although the high group tended to reach lower values, medians were still above the initially defined threshold of 0.800 G/L. Interestingly, this is in line with other studies on PICS, which have also shown very similar values, so the sensitivity of the propagated threshold may be questionable [18,21,42]. Readily identifying profuse catabolism can also be challenging. Serum albumin concentration was generally low and did not differ between the groups. Furthermore, its quality as a biomarker for catabolism is highly debatable in a surgical ICU setting [44,45]. The use of the creatinine height index is directly dependent on creatinine clearance and relies on expected creatinine excretion obtained from healthy subjects, which is not the case in an ICU setting. Here also, measuring weight loss is not always feasible, not to mention reliable. In order to estimate protein metabolism more precisely, we referred to the urea generation rate—measured as the sum of renal urea excretion and serum urea changes (Table 1). It must be mentioned that an elevated ureagenesis can be caused by degradation of endogenous or exogenously delivered protein. Although differentiation between the two cannot be made, an estimation of total metabolized protein can be deduced. Assuming that an average mix of amino acids has a nitrogen content of approximately 16%, back calculation from urea over nitrogen to protein is possible. The other surrogate used in this study is the serum urea-to-creatinine ratio, which has been described to predict catabolism in major trauma patients [39]. When compared to baseline or patients dismissed sooner, those with prolonged lengths of stay showed a significant increase in this ratio (>75 (mg/dL:mg/dL)). The pathophysiology behind the ratio is theoretically quite simple: muscle degradation and concomitant decline in muscle mass leads to elevated urea production but reduced creatinine generation [46]. Because the excretion of both rely on renal function, the ratio remains stable, fairly independent of acute kidney injury. Our study population with prolonged ICU LOS also had a clearly increased urea-to-creatinine ratio during the main observation period. Furthermore, the ratio significantly differed between the groups, discriminating the high group to be more catabolic than the low group. This fits in perfectly with a clear elevation of the urea generation rate in the high group when compared to the low group. While one must assume that an elevated urea generation rate is also associated with an elevated protein breakdown and hence catabolism, whether or not worse marker levels for inflammation (e.g., CRP) or immunosuppression (e.g., lymphocytes) are even associated with severity of illness regarding PICS remains debatable.

A link between PICS and hypernatremia can be established in two ways. On the one hand, hyperosmolar cell stress has been shown to trigger protein metabolism, muscle degradation and immunomodulation [32,33,34,35], and on the other hand, excessive catabolism has been shown to cause hypernatremia via profuse urine urea excretion, leading to osmotic diuresis and concomitant loss of electrolyte-free water [26,27,28,29,30,31]. Despite conceptualizing a potential vicious cycle, of course, the complex physiology behind PICS and hypernatremia cannot be broken down to one simple cause.

### 4.3. Assignable Causes of Hypernatremia

The genesis of ICU-acquired hypernatremia is always a result of loss of free water or gain of sodium [1,3]. As expected, initial sodium burden—mainly through resuscitation fluid—was generally high and even a bit higher in the high group, as shown in Figure 2. Following this peak, the median sodium balances were negative, even without consideration of extrarenal sodium losses. A slightly higher sodium input in the high group during ICU day 7–21 is attributable to the also higher fluid balance. Whether this is an attempt to treat or a possible cause of hypernatremia cannot be said at this point. Concentrating on free water losses, the EFWC has been proposed for discrimination between renal (EFWC positive) or extrarenal (EFWC negative) origin [1]. Extrarenal water losses include insensible losses, diarrhea or other losses (e.g., drains, chest tubes, nasogastric suction), which are predominantly included in daily fluid balances. Causes of renal water losses are further divided into osmotic diuresis (mainly from glucose or urea), the use of loop diuretics, renal insufficiency or diabetes insipidus. The latter two can be excluded by sufficient creatinine and free water clearances in our study population. In addition, glucose control is usually fairly tight via intravenous insulin in an ICU setting. Due to missing data, the use of loop diuretics cannot be excluded and is of unknown amount in this study. Its contribution to the observed hypernatremia cannot be clearly estimated. Although loop diuretics can decrease urine sodium concentration and, of course, induce polyuria, they usually will not raise the ratio of urea to other osmoles in renal osmole excretion, as shown in the high group. We conclude that the combination of this elevated ratio together with the measured polyuria and the reduced urine electrolyte concentration is what defines osmotic diuresis, leading to electrolyte-free water clearance and, in the end, at least partially contributing to the observed hypernatremia in the high group. Because the pathophysiology of catabolism inducing hypernatremia is often not thought of, these findings are of particular interest. Dealing with patients suffering from ICU-acquired hypernatremia while undergoing a prolonged ICU stay should at least trigger thoughts on possible ongoing catabolism and PICS as contributing factors.

### 4.4. Limitations

The main limitation of this study is the retrospective design within one single center. Centre-specific characteristics cannot be excluded. Missing data on the use of loop diuretics and scarce measurements of absolute lymphocytes is also unfortunate. As the criteria for chronic kidney disease were not met in our setting, we decided to set the threshold for sufficient renal function based on clinical thoughts regarding nonrequirement of renal replacement therapy, sufficient osmole excretion and ability to concentrate. As this threshold may therefore deviate from other studies, comparability may be hindered. Nevertheless, both the association between hypernatremia and PICS as well as the EFWC as a possible contributor to hypernatremia are presented well.

## 5. Conclusions

In ICU patients with a prolonged length of stay but fairly sufficient renal function, clinical signs of PICS are often present. A respectable number additionally suffer from ICU-acquired hypernatremia. With excessive catabolism present, one assignable cause for hypernatremia is urea-induced osmotic diuresis, subsequently causing loss of electrolyte-free water. When dealing with critically ill patients requiring prolonged ICU treatment, hypernatremia under sufficient renal function should at least trigger thoughts on catabolism and PICS as contributing factors.

## Figures and Tables

**Figure 1 jcm-09-03017-f001:**
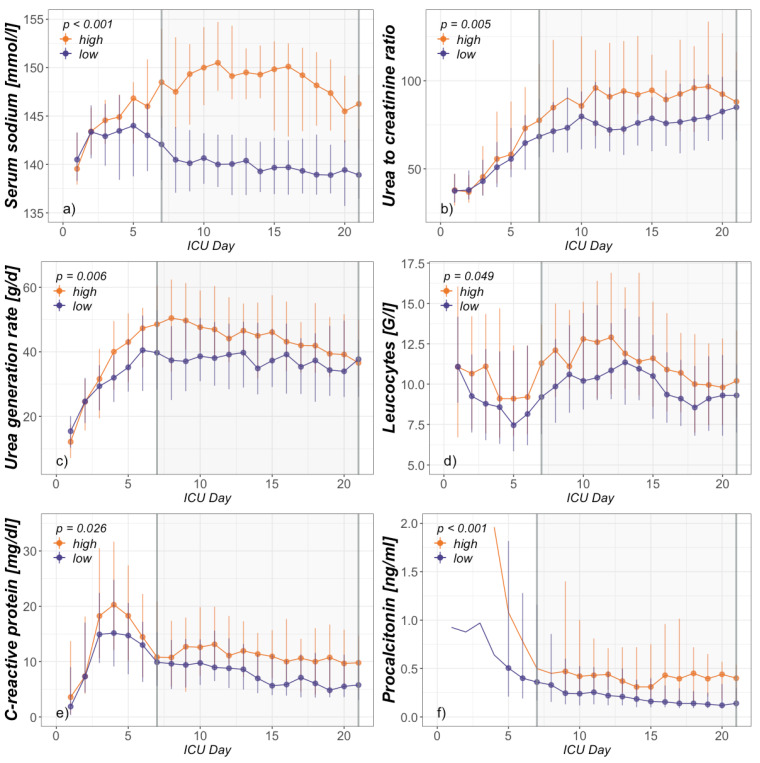
Hypernatremia, catabolism and inflammation. Intensive care unit (ICU) day-dependent group analysis presented as daily median and interquartile range from admission to day 21 of (**a**) serum sodium, (**b**) urea-to-creatinine ratio, (**c**) urea generation rate, (**d**) leucocytes, (**e**) C-reactive protein (CRP) and (**f**) procalcitonin. The range of statistical interest (ICU day 7–21) is indicated by the gray-shaded area.

**Figure 2 jcm-09-03017-f002:**
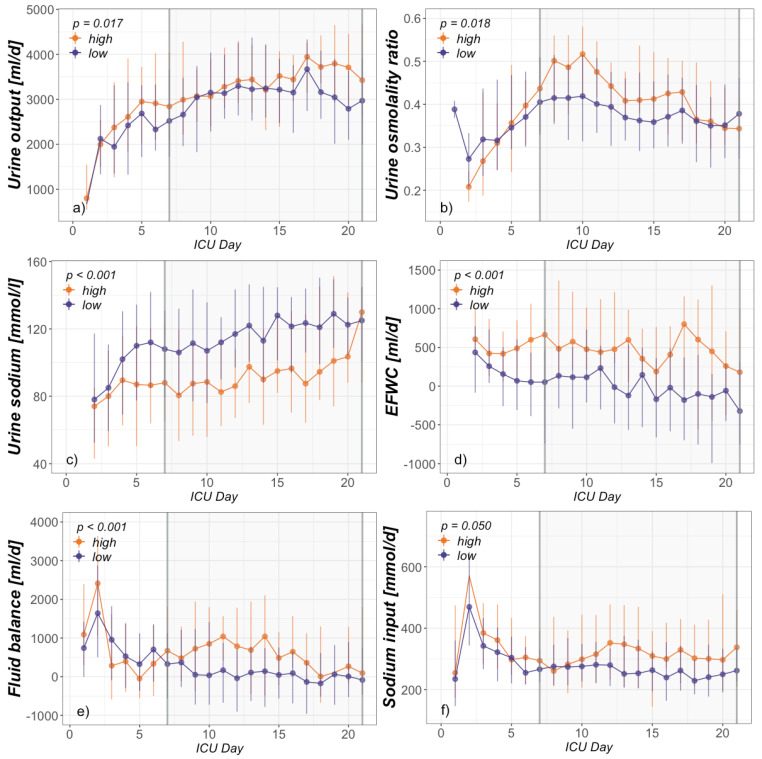
Electrolyte-free water clearance, fluid balance and sodium intake. ICU day-dependent group analysis presented as daily median and interquartile range from admission to day 21 of (**a**) urine output, (**b**) urine osmolality ratio (urea/not urea), (**c**) urine sodium output, (**d**) electrolyte-free water clearance (EFWC), (**e**) fluid balance and (**f**) sodium intake. The range of statistical interest (ICU day 7–21) is indicated by the gray-shaded area.

**Table 1 jcm-09-03017-t001:** Formulas.

Formula	Units
$FWC=OUT_{UR}\times \left[1-\left(\frac{Osmo_{Ur}}{Osmo_{P}}\right)\right]$	[mL/d]
$EFWC=OUT_{UR}\times \left[1-\left(\frac{1.03\times (Na_{Ur}+K_{Ur})}{Na_{S}+23.8+0.016\times \left(Gluc_{S}-120\right)}\right)\right]$	[mL/d]
$UreaGR=OUT_{UR}\times Urea_{Ur}+\left(Urea_{S\left[today\right]}-Urea_{S\left[yesterd.\right]}\right)\times 0.6\times body\;weight$	[g/d]

EFWC: electrolyte free water clearance; FWC: free water clearance; Gluc_S_: serum glucose concentration; K_Ur_: urine potassium concentration; Na_S_: serum sodium concentration; Na_Ur_: urine sodium concentration; Osmo_p_: plasma osmolality; Osmo_Ur_: urine osmolality; OUT_UR_: urine output; Urea_S_: serum urea concentration; Urea_Ur_: urine urea concentration; UreaGR: urea generation rate.

**Table 2 jcm-09-03017-t002:** Demographics.

Group	High (*n* = 33) Median (Q1–Q3) or *n* (%)	Low (*n* = 82) Median (Q1–Q3) or *n* (%)	*p*-Value
Age (years)	62 (49–68)	58 (44–69)	0.481
Gender			0.672
Female	10 (30.3%)	30 (36.6%)	
Male	23 (69.7%)	52 (63.4%)	
SAPS III *	66 (56–76)	69 (59–75)	0.538
Admission type			
Elective surgery	11 (33.3%)	15 (18.3%)	0.225
Emergency Surgery	5 (15.2%)	7 (8.5%)	
Trauma	10 (30.3%)	42 (51.2%)	
Sepsis	5 (15.2%)	12 (14.6%)	
Other medical	2 (6.1%)	6 (7.3%)	
ICU ^+^ length of stay (days)	23 (19–35)	19 (16–26)	0.030
Outcome			
Dead	2 (6.1%)	5 (6.1%)	1.000
Alive	31 (93.9%)	77 (93.9%)	

Data presented as median and interquartile range (IQR). *p*-Value refers to group differences for high vs. low. * Simplified Acute Physiology Score III [36,37] ^+^ Intensive care unit.

**Table 3 jcm-09-03017-t003:** Study characteristics during the main observation period (ICU day 7–21).

Group	Total Pop. (*n* = 115) Median (Q1–Q3)	High (*n* = 33) Median (Q1–Q3)	Low (*n* = 82) Median (Q1–Q3)	*p*-Value
Leucocytes (G/L)	10.4 (8.1–11.4)	11.2 (8.6–14.9)	10.0 (7.8–13)	0.049
Lymphocytes (G/L) *	1.078 (0.859–1.605)	0.933 (0.780–1.230)	1.250 (0.940–1.718)	0.080
CRP (mg/dL)	8.7 (4.7–14.2)	11.1 (6.1–17.8)	7.8 (4.4–13.1)	0.026
Procalcitonin (ng/mL)	0.26 (0.13–0.60)	0.42 (0.22–0.80)	0.20 (0.10–0.45)	<0.001
Albumin (g/dL)	1.71 (1.55–2.00)	1.72 (1.55–2.00)	1.71 (1.55–1.99)	0.962
Urea-to-creatinine ratio	79.6 (63.3–102.3)	91.0 (72.5–124.1)	75.3 (60.6–96.7)	0.005
Urea generation (g/d)	39.7 (29.4–51.8)	45.5 (35.5–57.7)	37.4 (27.5–48.7)	0.006
Renal protein loss (g/d)	118 (87–154)	133 (101–174)	111 (81–144)	0.007
Creatinine clearance (mL/min)	105.7 (71.2–158.3)	87.8 (62.2–127.3)	116.0 (78.1–172.2)	0.006
Free water clearance (mL/d)	−2276 (−3362–−1345)	−2095 (−3038–−1323)	−2378 (−3543–−1357)	0.161
Fluid balance (mL/d)	206 (−546–1096)	671 (−189–1685)	68 (−654–876)	<0.001
Sodium intake (mmol/d)	275 (205–382)	333 (208–440)	262 (205–352)	0.050
Serum sodium (mmol/L)	142 (138–147)	149 (144–153)	140 (137–143)	<0.001

Data presented as median and IQR. *p*-Value refers to group differences for high vs. low. ***** Due to scarcity of measurements, absolute lymphocyte count is compared as only one average value per patient in the timeframe ICU day 7–21.

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
