# Peer review of "ICU-Acquired Hypernatremia Is Associated with Persistent Inflammation, Immunosuppression and Catabolism Syndrome"

_jcm, 2020, doi:10.3390/jcm9093017_

Round 1
Reviewer 1 Report
The manuscript is much clearer and improved. This is especialy true for the conclusion.
Author Response
Dear Reviewer,
we are truly pleased our manuscript appeals to you!
With not much more to say, we would like to dearly thank you for your guidance and your helpfull suggestions in this review process!
With kind regards
Christopher Rugg
Reviewer 2 Report
The authors have addressed the concerns I spelled out in my original review and I thank them for that. Along with their edits based on other reviewers' suggestions, this is now a stronger paper and I recommend acceptance at this time in its present form.
Author Response
Dear Reviewer,
we truly appreciate your favourable comments. We are glad our modifications appeal to you!
At this point we would like to dearly thank you for accompaning us through this review process!
With kind regards
Christopher Rugg
This manuscript is a resubmission of an earlier submission. The following is a list of the peer review reports and author responses from that submission.
Round 1
Reviewer 1 Report
The most logical abbreviation for persistent inflammation, immunosuppression and catabolism syndrome (PICS) should be PIICS if I count the I’s. (Ref: Intensive Care Med (2020) 46:437–443. https://doi.org/10.1007/s00134-019-05851-3 Please reconsider.
This article focusses mainly on the catabolism aspect of chronic critically ill patients and not on the immunological part of the PIICS. The group of authors are complimented with their extensive collection of 24 hours urine in the ICU.
The authors state that sufficient renal function corresponds to 45 ml/min. Please provide reference for this chosen number.
Surrogate parameters for ongoing catabolism should include weight loss > 10% or BMI < 18 (ref Gentile 2012). This information is missing in the demographics table as is the gradient during ICU stay. If weight on admission or during ICU stay was not available in the PDMS maybe the authors can postulate on the weight loss using creatinin and or urea excretion in the 24 hours urine as marker for muscle loss.
Alinea 3.3: Assignable causes for hypernatremia; this was not mentioned in the study question / hypothesis. Figure 2 shows electrolyte free water clearance, fluid balance and sodium intake; lovely figures but it does not add in the association between hypernatremia and PIICs as for as I can see. Please make this more clear or defer to another manuscript.
Study question is : "We therefore hypothesize that in patients with prolonged ICU LOS but sufficient renal function hypernatremia is associated with PICS via chronic inflammation and excessive catabolism" this is not answered in the conclusion which is lined up as a mini abstract.
Is it possible to make a multivariate model with al the different parameters for PIICS and show the role of ICU acquired hypernatremia
Reviewer 2 Report
Hypernatremia is fairly frequent, but often overlooked. Its physiopathology is rarely discussed. This paper adds to the discussion of the causes of hypernatremia in critical patients, and its association with catabolism and inflammation. It is a welcome addition to the bibliography on the subject.
The only comment I would like to make is that the mechanism by which urea interferes with free water nephron reabsorption might be more complicated than osmotic diuresis - since, after all, urea is not an active osmol. In fact, urea could interfere with the normal function of Aquaporin 2 channels in the collecting duct, if studies done on toad bladder are indeed relevant. Only time will tell. However, since traditionally urea IS considered to interfere with water reabsorption in both the proximal and distal nephron by osmotic diuresis, and there is a dirth of bibliography on alternate urea actions, I am NOT suggesting that you modify what you have written, NOR incorporate my thoughts into your paper. In any event, urea's renal mechanism of action does NOT change your findings nor does it influence your conclusions.
Reviewer 3 Report
I read with interest the manuscript entitled “ICU-acquired hypernatremia is associated with persistent inflammation, immunosuppression and catabolism syndrome”. This is a retrospective study that evaluated adult patients for association between hypernatremia and PICS in ICU patients with a longer length of stay.
This is a well written manuscript that describes a new view of a well described and interesting problem in ICU patients. In short, I think the authors have done a solid job and made fairly compelling arguments for their conclusions. I would recommend that this manuscript be accepted, but with some revisions.
In terms of concerns and issues I have with the manuscript in its current form, my only major concern has to do with their basic findings. The authors note that in their final study population, all patients met criteria for PICS. Of these patients, the patients that also had hypernatremia simultaneously were found to have elevated markers of inflammation including urea generation, CRP, procalcitonin, etc. So, my question has to do with the significance of worse marker level in PICS. In other words, are elevated CRP, urea generation, etc. linked to worse outcomes? It seems from the data presented here that there was no difference in mortality but a difference in ICU LOS. That makes intuitive sense, but I need the authors to improve their explanation of the clinical significance of this difference in measured marker levels.
As far as other issues I see, they are relatively minor. I will list them below.
It would be very useful to include a sentence or two describing what the basic patient population of the ICU studied includes. Surgical? Medical? Mixed? All surgical specialties? Were neuro or neuro trauma patients included? Burns (they briefly mention in the discussion)?
The appendix figure showing the path for inclusion in the study is useful. As I was reading, I felt like it would have been helpful to have a clearer text explanation of how many patients were initially screened. The authors mention some numbers here regarding number excluded. It would be better to have all the numbers spelled out here (eg 1374 screened for eligibility) with the chart referred to for a visual representation of what is described.
In the discussion, on line 199, the authors note “negative outcome”. I think this is referring to mortality and, if so, the authors should state that explicitly. There are many negative outcomes not involving death, so the authors need to be clear what they are referring to here.
Earlier in the Demographic section of the Discussion, the authors talk about the ranges of reported ICU acquired hypernatremia. Another possible reason for the reported difference between this study and historical rates is the study inclusion criteria with longer LOS. If these other studies also only included longer LOS patients then they are correct, but this study is skewed towards longer LOS by its nature. And it would seem that hypernatremia is more related to increased LOS. The authors should expand on this and make it clearer.
In conclusion, this is very well written manuscript that raises important points for adult ICU practitioners. The data is presented clearly, and the conclusions fit the data without overstating the findings. I applaud the authors for this and NOT drawing a causal relationship here. I would hope that this retrospective work will lead to more prospective evaluation of this phenomenon which will allow for better control over variables and to more generalizable findings. Along with that, my final request would be for the authors to remove the words, at least from line 288 in the conclusion. I think the authors have made a strong enough argument that practitioners indeed should consider PICS in the setting of ICU acquired hypernatremia.